# Multi-Source Deep Transfer Neural Network Algorithm

**DOI:** 10.3390/s19183992

**Published:** 2019-09-16

**Authors:** Jingmei Li, Weifei Wu, Di Xue, Peng Gao

**Affiliations:** College of Computer Science and Technology, Harbin Engineering University, No.145 Nantong Street, Harbin 150001, China; lijingmei@hrbeu.edu.cn (J.L.); dixue@hrbeu.edu.cn (D.X.); gaopeng1979@hrbeu.edu.cn (P.G.)

**Keywords:** multi-source transfer learning, deep learning, convolutional neural network, classification

## Abstract

Transfer learning can enhance classification performance of a target domain with insufficient training data by utilizing knowledge relating to the target domain from source domain. Nowadays, it is common to see two or more source domains available for knowledge transfer, which can improve performance of learning tasks in the target domain. However, the classification performance of the target domain decreases due to mismatching of probability distribution. Recent studies have shown that deep learning can build deep structures by extracting more effective features to resist the mismatching. In this paper, we propose a new multi-source deep transfer neural network algorithm, MultiDTNN, based on convolutional neural network and multi-source transfer learning. In MultiDTNN, joint probability distribution adaptation (JPDA) is used for reducing the mismatching between source and target domains to enhance features transferability of the source domain in deep neural networks. Then, the convolutional neural network is trained by utilizing the datasets of each source and target domain to obtain a set of classifiers. Finally, the designed selection strategy selects classifier with the smallest classification error on the target domain from the set to assemble the MultiDTNN framework. The effectiveness of the proposed MultiDTNN is verified by comparing it with other state-of-the-art deep transfer learning on three datasets.

## 1. Introduction

In the past two decades, machine learning has dramatically progressed, and it has become a practical technology from laboratory to widespread commercial use [1]. Currently, machine learning is one of the fastest growing technologies located at the core of artificial intelligence and data science, which has been widely used in intrusion detection [2,3], speech recognition [4,5], computer vision [6,7], spam detection [8,9], pattern recognition [10], text classification [11], and other fields. Of course, it has achieved great results. However, in order to obtain a high accuracy classification model, many machine learning algorithms need to satisfy the following two basic conditions: (1) the training and test data come from the same feature space and the same distribution, which satisfy the independent and identical distribution conditions; (2) enough training samples are available. Nevertheless, these assumptions are not always met in practical applications [11,12]. Especially in emerging applications such as text mining, bioinformatics, distributed network sensor networks, and social network research, the independent and identical distribution conditions cannot be satisfied between training and test data under the influences of time, environmental changes, or instability of sensor devices. When the data distribution changes, most of the models need to re-collect the training data, but the previous training data will not be used again, and this results in wasted data resources. In addition, data sample resources in some areas are often scarce, and the cost of collecting data is very expensive or even impossible. In this case, knowledge transfer between task domains is desirable [12,13].

Transfer learning, also known as domain adaptation, provides an effective means to solve the above problems. On the one hand, it no longer requires training and test data to satisfy independent and identical distribution conditions. On the other hand, when the training data in the target domain is scarce and not enough to obtain a good classifier, the data from the source domain (often containing a large number of labeling samples) is similar to the target domain and can be used to assist the learning tasks in the target domain. Transfer learning has achieved remarkable results in resisting this challenge by transferring knowledge from source to target domains with different distributions [13]. Therefore, transfer learning attracts more and more researcher attention and has made great progress: Gao et al. [14] proposed a local weighted embedded transfer learning algorithm LWE; a feature-based space transfer learning method LMPROJ are proposed by Brain et al. [15]; Lu et al. [16] proposed a selective transfer algorithm STLCF for collaborative filtering; Long et al. [17] proposed an SVM-based least squares transfer learning framework ARTL; Xie et al. [18] applied transfer learning to incremental learning and proposed an STIL algorithm; Li et al. [19] proposed a new transfer learning algorithm TL-DAKELM based on the extreme learning machine; Li et al. [20] proposed a transfer learning algorithm, RankRE-TL.

The above transfer algorithms can only handle a single source domain, but in many real-world applications, data from more than one source domain can be collected. Therefore, transfer learning algorithms with multi-source domains are naturally researched, and the classification effect is better than that using only one source domain [21]. Recently, machine learning algorithms of transfer learning with multi-source domains have been proposed. Yao et al. [22] extended the boosting framework and proposed MultiSource-TrAdaBoost and TaskTrBoost; Sun et al. [23] proposed a two-stage domain adaptive method that combines weights of data on marginal probability differences (first phase) and conditional probability differences (second phase) from multiple source and target domains; Duan et al. [24] proposed a multi-source domains adaptation method DAM; [25] proposed a new online transfer learning algorithm by using labeling data from multiple source domains to seek to improve classification performance in target domain; Ding et al. [26] attempted to use the incomplete multi-source domains to carry out effective knowledge transfer, and proposed an incomplete multi-source transfer learning to improve knowledge transfer in two directions; In [27], Jun et al. explored two problems of domain adaptation and proposed the A-SVM algorithm.

No matter multi-source or single-source transfer learning algorithms, although the classification effect of the above transfer learning algorithms can be accepted, in fact these algorithms belong to a shallow structure. Therefore, they cannot find deeper and more complex knowledge behind the data, and then find more common information between domains to further improve the classification effect in target domain.

With the emergence of deep learning, it has more powerful expression ability than the learning algorithms of shallow structure, and has gained a lot of attention for the advantage of better representation features. Consequently, deep learning can generate more domain-invariant features for knowledge transfer between domains. At present, there are many research works on the combination of transfer learning and deep learning. Huang et al. [28] proposed a shared hidden layer multilingual DNN (SHL-MDNN), in which the hidden layer is common in many languages, while the softmax layer is language dependent. Ding et al. [29] developed a new deep transfer low- rank coding based on a deep convolutional neural network, which can obtain a multi-layer general dictionary shared across two domains to bridge domain gaps, so that rich domain invariant knowledge can be captured by the way of layering. The deep transfer learning framework was proposed by Te et al. [30] extended marginal distribution adaptation to joint distribution adaptation and uses unambiguous structures associated with labeled samples of source domain to adjust the conditional distribution of the unlabeled samples in target domain, which ensures a more accurate distribution matching. [31] proposed a new deep adaptive network architecture Domain Adaptation Network (DAN), which extended the deep convolutional neural network to the domain adaptation scenario, the architecture learns the transferable features through statistical guarantees and can be embedded through the kernel without bias, and is estimated to perform linear expansion. A CNN framework that utilizes unlabeled or sparsely labeled data in the target domain is proposed to facilitate transfer by optimizing domain invariance [32]. Zhang et al. [33] proposed a new method for deep convolutional neural networks, Deep Convolutional Neural Networks with Wide First layer Kernels (WDCNN) that uses the original vibration signal as input and wide kernel in the first convolutional layer to extract features and suppress high frequencies noise. The proposed DHN algorithm aims to seek informational hash coding by combining deep structure learning with domain alignment [34]. DDC is the first to incorporate domain aliasing losses into the top layer of AlexNet to transfer drift during domain transfer [35]. But these algorithms only consider the differences of marginal probabilities distribution in domains and the knowledge from only the single source domain, and ignore conditional probabilities and intrinsic information of domains.

In this paper, inspired by previous researches on the combination of deep neural networks and transfer learning, we propose a new multi-source deep transfer neural network algorithm (MultiDTNN) based on convolutional neural networks and multi-source transfer learning. The core idea of this work is as follows: First, to enhance the feature transferability in specific layers in deep neural networks by reducing the domain differences between each source and target domain with using joint probability distribution adaptation (JPDA). Then, we train Convolutional Neural Networks (CNN) on each source and target domain to get a set of classifiers. Finally, for the sake of gaining MultiDTNN, the second stage of the TaskTrAdaBoost [22] algorithm is applied to design a selection strategy to select the classifier with the smallest classification error on target domain from the classifier set. To the best of our knowledge, we are the first to apply multi-source transfer learning and JPDA to the classification tasks of cross-domain knowledge transfer on deep neural networks.

Our threefold contributions are highlighted as follows: (1) the deep transfer structures are constructed based on JPDA and a convolutional neural network which can transfer more features of data in the source domain to the target domain; (2) more knowledge in multi-source domains are provided to assist in building the learning model of target domain, so the classification effect of the model is better; (3) ensemble system of classifiers is more advantageous than a single classifier in terms of prediction effectiveness and stability.

The remaining parts of the paper are organized as follows: In Section 2, the related works of multi-source transfer learning, convolutional neural networks, and maximum mean discrepancy are briefly discussed. The MultiDTNN is proposed and implementation details are also explained in Section 3. Section 4 verifies the effectiveness of MultiDTNN by comparing with state-of-the-art benchmark algorithms on three cross domain datasets. The last section summarizes the conclusions of this paper.

## 2. Related Works

### 2.1. Multi-Source Transfer Learning

Transfer learning has been extensively studied for many years since it was proposed in NIPS-95 in 1995 [12]. However, in real-world applications, we can easily collect auxiliary data from multiple source domains. Therefore, the studies of multi-source domains transfer learning have gradually attracted the interest of researchers [13,14,15,16,17,18,19,20,21,22,23,24,25]. It can transfer knowledge from multiple source domains to learning tasks of the target domain compared to previous transfer learning algorithms with single domains [26]. In addition, if there is no or weak correlations between target and source domains, transfer learning not only has no ability to improve the performance of the target domain classifier, but also lead to negative transfer, on the contrary which will reduce the performance of target domain classifier. Therefore, when extracting knowledge from two or more source domains, the knowledge of data in source domains with more closely related to target domain is selected as much as possible to create a prediction model in target domain [27]. As shown in Figure 1, multi-source transfer learning makes use of the relationships between multi-source and target domains to improve the prediction performance of the samples in target domain, and assists in target domains to establish a prediction model.

In Figure 1, $\left(D_{S_{1}},T_{S_{1}}\right)$, $\left(D_{S_{2}},T_{S_{2}}\right)$, …, $\left(D_{S_{n}},T_{S_{n}}\right)$ respectively represent source domains and corresponding learning tasks. Similarly, $\left(D_{T},T_{T}\right)$ is target domain and corresponding learning tasks. $f_{t}$ denotes classifier that is obtained by the way of training transfer learning system with using the datasets in target and source domains.

Multi-source transfer learning can be divided into two categories: the boosting-based methods [22,25] and regularization-based methods [26,27]. The regularization-based methods are the learning model with the regularization term to solve the optimization problems, and the boosting-based methods use the boosting algorithm to generate the set of classifiers. In this paper, the proposed MultiDTNN belongs to the latter. While multiple source domains can provide more knowledge, the differences of domains also present challenging transfer learning issues. To this end, many methods for solving the schemes of multi-source domains have been proposed in many practical applications [22,23,24,25,26,27].

### 2.2. Convolutional Neural Network

In the past few years, deep learning has achieved good performance in solving various problems. CNN has been extensively studied in different types of deep neural networks [36]. In 2006, Hinton et al. published a paper on Science, which first proposed a convolutional neural network [37]. As one of the most effective deep learning models, CNN has been widely used in image processing [38,39,40], face recognition [41] and feature extraction [42]. In general, a CNN consists of three parts: convolutional layers, pooling layers, and fully connected layers. The convolutional layer and the pooling layer are alternately arranged; that is, one convolutional layer is followed by one pooling layer, and so on. After the multiple convolutional and pooling layers, one or more fully connected layers are connected. The first step in CCN convolves the input signal to obtain a feature map through the use of convolution kernel, and then uses a nonlinear activation function (ReLU) to act on the feature map. The formal description of the convolution layer operation is as follows:(1)$$c_{n}^{r}=\mathrm{ReLU}(\sum_{m}v_{m}^{r-1}*w_{n}^{r}+b_{n}^{r})$$

In Equation (1), $c_{n}^{r}$ is the n-th output of convolutional layer r, n denotes the number of convolution kernels in convolutional layer r, $w_{n}^{r}$ and $b_{n}^{r}$ respectively represent the convolutional kernel and the deviation, $v_{m}^{r-1}$ is the m-th output of convolutional layer r−1, ∗ is the convolutional operation. After calculating Equation (1), we can obtain the feature map and then perform average or maximum feature activation through the pooling layer in areas where the feature map does not intersect. Finally, the fully connected layer is used for classification. Given a data set $X={\left\{x_{i}\right\}}_{i=1}^{M}$, a CNN optimization learning process with P convolutional layers, a convolution kernel parameter set ${\left\{W_{i}\right\}}_{i=1}^{P}$, a bias set ${\left\{b_{i}\right\}}_{i=1}^{k}$ and a fully connected layers ${\left\{b_{i}\right\}}_{i=1}^{k}$ can be defined as:(2)$$\underset{{\left\{W_{i}\right\}}_{i}^{P},{\left\{b_{i}\right\}}_{i}^{P},U}{\min}\sum_{j}l(Y(x_{j}),f(x_{j},{\left\{W_{i}\right\}}_{i}^{P},{\left\{b_{i}\right\}}_{i}^{P},U))$$
where l(⋅) denotes the loss function to estimate the cost between true label $Y(x_{j})$ and predicted label by CNN model $f(x_{j},{\left\{W_{i}\right\}}_{i}^{P},{\left\{b_{i}\right\}}_{i}^{P},U)$.

### 2.3. Maximum Mean Discrepancy

Since the proposed MultiDTNN needs to measure the distribution differences between domains, it is necessary to choose a suitable measurement method of distribution distance. It has recently been demonstrated that the maximum mean deviation (MMD) in the regenerative kernel Hilbert space is a valid method for estimating the distance between two distributions [43]. For the convenience of calculation, the square form of MMD is generally used. The process of estimating the difference between two domains using MMD is as follows. 

Given a labeled dataset in a source domain $D_{s}=(\{x_{1},y_{1}\},\ldots ,(x_{n},y_{n}))$, an unlabeled dataset in target domain $D_{t}=(z_{1},\ldots ,z_{m})$, the nonlinear mapping function in the regenerative kernel Hilbert space is ϕ. The squared form of MMD is defined as follows:(3)$$MMD_{H}^{2}=||\frac{1}{n}\sum_{i=1}^{n}\phi (x_{i})-\frac{1}{m}\sum_{i=1}^{m}\phi (z_{i})|{|}^{2}$$

In Equation (3), the differences of distribution between two domains is the distance between the two data distributions. The smaller of MMD value, the closer the two domains are. If the value is 0, the two domains match. At present, MMD have been widely used in transfer learning algorithms [15,21,23,24,26,29,30,32], which can be used to construct regularization terms to learn features in different domains with more similar. In neural network-based transfer learning algorithms, MMD is often added to the loss function for optimization [30].

## 3. Multi-Source Deep Transfer Neural Network

This section describes the multi-source deep transfer neural network algorithm in detail. For convenience, we only consider the binary classification problem. Given N source domains are defined as:$D_{s}=\{D_{s_{i}}={\left(x_{j}^{s_{i}},y_{j}^{s_{i}}\right)}_{j=1}^{n_{s_{i}}},i=1,\ldots ,N\}$, $x_{j}^{s_{i}}$ denotes j-th sample of $s_{i}-th$ source domain, the corresponding class label is $y_{j}^{s_{i}}$, $n_{s_{i}}$ is the number of sample in $s_{i}-th$ source domain, $P_{s_{i}}$ and $Q_{s_{i}}$ mean marginal and conditional probability distribution. Analogously, target domain is $D_{T}=(x_{i}){|}_{i=1}^{n_{t}}$, marginal and conditional probability distribution are $P_{t}$ and $Q_{t}$. Normally, $P_{s_{i}}\neq P_{t}$ and $Q_{s_{i}}\neq Q_{t}$. 

In this paper, the goal of our proposed MultiDTNN is to use knowledge from multi-source domains to assist learning tasks of target domains to create an efficient classifier model, which can accurately label unlabeled samples in target domains. In MultiDTNN, knowledge transfer from the source to target domains is achieved through transfer learning [11]. Transfer learning is a new machine learning that solves learning problems in different but related domain (target domain) by using knowledge in existing historical data (source domain) [44,45]. At present, most of the transfer learning techniques commonly used by researchers are instance-based methods, which select representative instances from source domain to assist learning tasks in target domain [22]. However, target and source domains differ greatly in practical applications, if the instance data of source domain that is not related to target domain are forcibly transferred to target domain, which will not help the learning of target domains named as negative transfer. The negative transfer has been born with transfer learning, and it has always been the focus of researchers. In order to avoid negative transfer and better assist the learning tasks in target domain, it is particularly important to select samples in source domain with high similarity to target domain [12,13]. MultiDTNN can transfer knowledge from multiple source domains into the target domain, so as to improve the classifier effect, and we must fully consider the difference between each source and target domains, maximizing the knowledge transfer from source domains similar to target domains to avoid negative transfer. The composition strategy, the knowledge transfer from multi-source domains, and the classifier training process in the MultiDTNN model are described in detail below.

### 3.1. Joint Probability Distribution Adaptation

In practical applications, each source and target domains are not only different in marginal probability, but also have significant differences in conditional probability. If only the marginal probability between the source and target domains is considered, the negative transfer phenomenon may occur, and the better classification performance cannot be achieved in transfer learning. Therefore, in order to make the proposed MultiDTNN a better classification effect, we simultaneously consider both the marginal and conditional probability. Literature [30,46] points out that minimizing the differences of marginal and conditional distributions can effectively avoid negative transfer and improve the classification performance of transfer learning algorithms.
(4)$$\min Diff(P_{s_{i}}(\phi (x^{s_{i}})),P_{t}(\phi (x^{t})))$$
(5)$$\min Diff(Q_{s_{i}}(y^{s_{i}}|(x^{s_{i}})),Q_{t}(y^{t}|\phi (x^{t})))$$

In Equations (4) and (5), ϕ(⋅) represents a feature mapping to a regenerating kernel Hilbert space, $x^{s_{i}}$ is sample vector and $y^{s_{i}}$ is label vector in $s_{i}-th$ source domain. $x^{t}$ is sample vector and $y^{t}$ is label vector in target domain. Diff represents a function that calculates the differences between the source and target domains.

Equation (4) is to minimize the data distribution distance between the target and source domains. We apply MMD (Equation (3)) to calculate Equation (4):(6)$$MMD_{H}^{2}(P_{s_{i}},P_{t})=||\frac{1}{n_{s_{i}}}\sum_{j=1}^{n_{s_{i}}}\phi (x_{i}^{s_{i}})-\frac{1}{n_{t}}\sum_{j=1}^{n_{t}}\phi (x_{i}^{t})|{|}_{H}^{2}$$

The conditional distribution in (5) is intractable because of unknown $y^{t}$. We rewrite it into the following Equation (7):(7)$$\min D(\frac{Q_{s_{i}}(\phi (x^{s_{i}})|y^{s_{i}})\cdot P_{s_{i}}(\phi (x^{s_{i}}))}{P_{s_{i}}(y^{s_{i}})},\frac{Q_{t}(\phi (x^{t})|y^{t})\cdot P_{t}(\phi (x^{t}))}{P(y^{t})})$$

In order to solve the problems of the unknown sample label of the target domain, the literature [30,31] proposed a circuitous way: Equation (7) is processed by using the pseudo labels of data in the target domain. That is, by means of the pre-training model on labeled source data, pseudo labels in target domain will be obtained. The calculation method of samples pseudo-label in target domain is as follows: the similarity weight of samples in source and target domain is preferably calculated by using the MMD method, then the CNN classifier is trained by using the samples in the source domain and corresponding weight information, and finally the samples pseudo-label in target domain are labeled by the classifier. Supposing a total of C categories in target domain, c∈{1,…,C}. We utilize Equation (3) to measure the mismatch of conditional distributions with $Q_{s_{i}}(x^{s_{i}}|y^{s_{i}}=c)$ and $Q_{t}(x^{t}|y^{t}=c)$:(8)$$MMD_{H}^{2}(Q_{s_{i}}^{\left(c\right)},Q_{t}^{\left(c\right)})=||\frac{1}{n_{s_{i}}^{\left(c\right)}}\sum_{x_{j}^{s_{i}}\in D_{s_{i}}^{\left(c\right)}}\phi (x_{j}^{s_{i}})-\frac{1}{n_{t}^{\left(c\right)}}\sum_{x_{j}^{t}\in D_{t}^{\left(c\right)}}\phi (x_{j}^{t})|{|}_{H}^{2}$$
where $D_{s_{i}}^{\left(c\right)}=\{x_{j}^{s_{i}}:x_{j}^{s_{i}}\in D_{s_{i}}\wedge y(x_{j}^{s_{i}})=c\}$, $y(x_{j}^{s_{i}})$ is the true label, and $n_{s}^{\left(c\right)}=|D_{s}^{\left(c\right)}|$, $D_{t}^{\left(c\right)}=\{x_{j}^{t}:x_{j}^{t}\in D_{t}\wedge y(x_{j}^{t})=c\}$, $y(x_{j}^{s_{i}})$ is the pseudo label and $n_{t}^{\left(c\right)}=|D_{t}^{\left(c\right)}|$. There are certainly many errors in the initial pseudo labels of target data, but we can iteratively update the pseudo labels in subsequent model optimization stages until the best prediction accuracy is obtained.
(9)$$D_{H}(J_{s_{i}},J_{t})=MMD_{H}^{2}(P_{s_{i}},P_{t})+\sum_{c=1}^{C}MMD_{H}^{2}(Q_{s_{i}}^{\left(c\right)},Q_{t}^{\left(c\right)}),$$

In Equation (9), $J_{s_{i}}$ and $J_{t}$ is the JPDA of $s_{i}-th$ source domain $D_{s_{i}}$ and target domain $D_{t}$. The minimization of Equation (9) ensures the match in marginal and conditional distributions with sufficient statics.

### 3.2. Construction of MultiDTNN

Based on JPDA in Section 3.1, we use convolutional neural network to establish a multi-source deep transfer neural network framework. The framework of MultiDTNN is shown in Figure 2.

From Figure 2, we divide the MultiDTNN into two parts: a set of classifier which contains N classifiers $TCNN_{s_{i}}$ is obtained by training on CNN with JPDA using source domain $D_{s_{i}}$ and a target domain; we implement a selectin strategy similar to that in [22] to choose ensemble of classifier, which composes the model of MultiDTNN. Ensemble is the system that uses multiple predictors, statistically independent to some extent, in order to attain an aggregated prediction [47]. Such systems usually perform better than a single predictor, and their stability is better. The two parts are described in detail below.

A. Construction of $TCNN_{s_{i}}$

The structure of $TCNN_{s_{i}}$ is shown in Figure 3. In general, we can train the CNN model on sufficient data in source domain from scratch by using the optimization task defined in Equation (2). When applying the pre-trained CNN model to the target domain, we integrate JPDA and as a loss function regularization term, redefining the new objective function as:(10)$$L(\theta )=l_{c}+\lambda D_{H}(J_{s_{i}},J_{t})$$

$\theta ={\left\{W^{i},b^{i}\right\}}_{i=1}^{l}$ is the parameter set of a CNN with l layers and λ is non-negative regularization term. For CNN, as the number of layers increases, the features will change from general to specific. The upper layer tends to represent more abstract features, which will lead to larger domain differences. Therefore, we deploy regularization operations on the fully connected layer.

By minimizing Equation (10), we can adapt the pre-trained CNN to the classification task of the target domain. We use a mini-batch stochastic gradient (SGD) [29,30] and a backpropagation algorithm for the optimization of CNN networks. The gradient of Equation (10) for network parameters is as follows:(11)$$\nabla_{\theta^{l}}=\frac{\partial l_{c}}{\partial_{\theta^{l}}}+\lambda {\left(\nabla D_{H}(J_{s_{i}},J_{t})\right)}^{T}(\frac{\partial \phi (x)}{\partial_{\theta^{l}}})$$

The detailed formations of $\nabla D_{H}(J_{s_{i}},J_{t})$ are described as:(12)$$\nabla D_{H}(J_{s_{i}},J_{t})=\nabla MMD_{H}^{2}(P_{s_{i}},P_{t})+\sum_{c=1}^{C}\nabla MMD_{H}^{2}(Q_{s_{i}}^{\left(c\right)},Q_{t}^{\left(c\right)})$$
(13)$$\nabla MMD_{H}^{2}(P_{s_{i}},P_{t})=\left\{ \begin{array}{l} \frac{2}{n_{s_{i}}}(\frac{1}{n_{s_{i}}}\sum_{j=1}^{n_{s_{i}}}\phi (x_{j}^{s_{i}})-\frac{1}{n_{t}}\sum_{j=1}^{n_{t}}\phi (x_{j}^{t})),x\in D_{s_{i}}\\ \frac{2}{n_{t}}(\frac{1}{n_{t}}\sum_{j=1}^{n_{t}}\phi (x_{j}^{t})-\frac{1}{n_{s_{i}}}\sum_{i=1}^{n_{s_{i}}}\phi (x_{i}^{s_{i}})),x\in D_{t} \end{array}\right.$$
(14)$$\nabla MMD_{H}^{2}(Q_{s_{i}}^{\left(c\right)},Q_{t}^{\left(c\right)})=\left\{ \begin{array}{l} \frac{2}{n_{s_{i}}^{\left(c\right)}}(\frac{1}{n_{s_{i}}^{\left(c\right)}}\sum_{x_{j}^{s_{i}}\in D_{s_{i}}^{\left(c\right)}}\phi (x_{j}^{s_{i}})-\frac{1}{n_{t}^{c}}\sum_{x_{j}^{t}\in D_{t}^{\left(c\right)}}\phi (x_{j}^{t})),x\in D_{s_{i}}\\ \frac{2}{n_{t}^{\left(c\right)}}(\frac{1}{n_{t}^{c}}\sum_{x_{j}^{t}\in D_{t}^{\left(c\right)}}\phi (x_{j}^{t})-\frac{1}{n_{s_{i}}^{\left(c\right)}}\sum_{x_{j}^{s_{i}}\in D_{s_{i}}^{\left(c\right)}}\phi (x_{j}^{s_{i}})),x\in D_{t} \end{array}\right.$$

The training procedure mainly consists of two subprocesses: (1) pre-trained CNN on each labeled source domain data; (2) network adaptation in target domain using labeled data of source domain data and unlabeled data of target data by training CNN classification of (1). Therefore, we can get a collection of classifiers $H\in {\left\{TCNN_{i}\right\}}_{i=1}^{N}$ on N source domains. The detailed procedure is shown Step 1 in Table 1. When the size of data in source domain becomes large, the calculation of CNN requires the support of high-performance computers, which is also the need for deep learning in the future. Therefore, in order to better record the performance indicators during the operation to provide support for optimizing CNN, various performance tuning tools are used.

B. Strategy of selection

In order to get a powerful set of classifiers, we are inspired by [22] to implement an efficient strategy of selection. The strategy is as follows: the AdaBoost algorithm is cyclically executed on dataset of target domain, and a classifier is selected from each of the classifier sets in each iteration, and the classifier is trained on target domain; ensure that the knowledge of source domain is more closely related to the target task is transferred, calculate the error rate of the classifier on target domain dataset, and select the classifier which the error rate meets the requirements, else discard the classifier; in addition, the weight of the sample of target domain is updated for the next iteration. The detailed selection process is shown Step 2 ~Step 13 in Table 1. In the end, we will get a set of classifiers with better classification performance on target domain, which is our proposed MultiDTNN model.

### 3.3. Training Strategy of MultiDTNN

According to Section 3.1, Section 3.2 and Section 3.3, the training process of proposed MultiDTNN is summarized and described in Table 1.

## 4. Experimental Results

In this section, in order to analyze the effectiveness of the proposed MultiDTNN, we evaluate it on three cross-domain standard datasets. First, the experimental setup is introduced in Section 4.1. Then, Section 4.2 describes the three cross-domain datasets in detail. Finally, in Section 4.3 we compare the proposed MultiDTNN with several state-of-the-art deep transfer learning algorithms. 

### 4.1. Experimental Setting

The following state-of-the-art transfer learning methods are chosen as benchmark algorithms for comparison with MultiDTNN: ARTL [12], STLCF [16], TaskTrBoost [22], FastDAM [24], IMTL [26], DTLC [29], DAN [31], SDT [32], DHN [34], DDC [35], CNN [38], and Deep CORAL [40]. Among these benchmark algorithms, CNN is a non-transfer learning algorithm, TaskTrBoost, FastDAM, and IMTL are transfer learning algorithms that can utilize knowledge in multiple source domains, STLCF and ARTL are non-deep transfer learning. For baseline methods, we adopt the standard procedures for model as described in their respective works to our paper. We implement the proposed MultiDTNN using TensorFlow and train with Stochastic Gradient Descent (SGD). The initial learning rate is set as $10^{-3}$, and momentum is 0.9 in SGD. The parameters λ is searched in the range from 0.01 to 100. Actually, MultiDTNN model can easily adopt other CNN structures, e.g., VGGNet, ResNet, and GoogleNet. Deeper CNN structures would improve the performance somehow. Since we are focusing on the specific layers, we only evaluate the AlexNet structure in this paper. We primarily follow an unsupervised standard evaluation protocol to adopt and use all labeled samples of source domain and unlabeled samples of target domain. For the fairness of experiments, a 5-fold cross-validation strategy is selected for all experiments, and we repeat the strategy twice as the final comparison results. In the experiments we will run 10 times, the average value of classification accuracy, with their standard deviations are recorded. The representation of classification accuracy is as follows:$$Accuracy=\frac{\left|x:x\in D_{t}\wedge f(x)=y(x)\right|}{\left|x:x\in D_{t}\right|}\times 100\%$$
where the dataset of target domain is $D_{t}$, y(x) represents the truth class label of x, f(x) is the class label of x predicted by the classifiers. 

### 4.2. Datasets

Office-31, Office-10+Caltech-10 and Office+Home [30,31,32] are commonly well-known cross-domain standard datasets in transfer learning applications, so all experiments in this paper are performed on these datasets. The datasets are described in detail below.

Office-31 is a standard dataset that contains 4,652 images from the domains Amazon (A), Webcam (W), and DSLR (D). These images can be divided into 31 categories. Among them, the samples in Amazon are from www.amazon.com, and the samples in Webcam and DSLR are obtained through web cameras and digital SLR cameras in different environments. We construct six cross-domain tasks A->D, A->W, W->A, W->D, D->A, and D->W from source to target domains. On each of the above-mentioned cross-domain, the proposed multi-source MultiDTNN algorithm uses A, W, and D as the source domain.

Office-10+Caltech-10 contains 10 common objects shared by Office-31 and Caltech-256 (C)^2^ datasets, which have been widely used in domain adaptation methods. As with the method of constructing cross-domain tasks on Office-31, we construct 12 cross-domain tasks. The number of source domain is 4 in MultiDTNN.

Office+Home collects objects from 4 domains: Art (Ar, artistic drawing object), Clipart (Cl, images collected from www.clipart.com), Product (Pr, similar to Amazon’s sample with almost clean background) and Real-World (Re, object images taken with regular camera). The dataset has 65 objects with15500 image samples. Similarly, we constructed 12 cross-domain tasks in a similar way to Office-31, with MultiDTNN using 4 source domains simultaneously on each task.

### 4.3. Analysis of Experimental Results

In this section, the experimental results of MultiDTNN algorithm and 12 benchmark algorithms on real datasets are analyzed and compared. We compare the average accuracy rate after 10 experiments on the three datasets. Table 2 shows the results of six cross-domain tasks on Office-31. The results of 12 cross-domain tasks on Office-10+Caltech-10 are shown in Table 3. Table 4 shows the results on 12 cross-domain tasks of Office+Home.

From the results in Table 2, Table 3 and Table 4, we can draw the following conclusions:

(1) On the cross-domain tasks of three datasets, the average accuracy rate of the based deep learning methods outperform the common transfer learning algorithms ARTL and STLCF, which shows that the based deep learning methods are obviously superior to the shallow transfer learning algorithm.

(2) CNN-based deep transfer learning algorithms (e.g., DAN, DTN, SDT, D-COREL, DTLC, and DHN) can use the knowledge of source domain to assist in learning tasks in target domain, so their classification performance is better than standard deep learning method (CNN). This indicates that the data in source domain can be used to improve the learning task of target domain with unlabeled data on the deep neural network model combined with transfer learning, so their experimental results are better.

(3) In the benchmark algorithms, TaskTrBoost, FastDAM, and IMTL can utilize the sample features of multiple source domains to help learning tasks of target domain create classifier models, so their classification effect is better than ARTL and STCF, which are non-deep single source domain transfer learning algorithms, and even are obviously superior to CNN-based deep transfer learning algorithms in some cases.

(4) Comparing with Office-31 and Office-10+Caltech-10, Office+Home contains more categories and the distribution between categories is larger, so all algorithms cannot achieve promising performance. However, from the experimental results we could notice that our proposed model obtain better performance in most cases. Especially in Office+Home, MultiDTNN can achieve better performance than the benchmark algorithms.

(5) Comparing with the benchmark algorithms, our proposed MultiDTNN model can transfer knowledge from more than one source domain, so it can help the learning tasks of the target domain to build a more efficient classifier model. For example, for cross-domain task A->W of the dataset Office-31, the transfer deep neural network algorithms DTLC, DAN, SDT, DHN, D-CORAL, and DDC of the benchmark algorithms can only transfer the knowledge of one source domain A to the target domain W. Nevertheless, the proposed MultiDTNN can simultaneously use the knowledge of three source domains A, W, and D for the learning task of the target domain. Similarly, the number of source domains that MultiDTNN can utilize on the Office-10+Caltech-10 and Office+Home datasets is 4. We carefully analyzed all the experimental results on the three datasets, and see that MultiDTNN works best. In addition, the experimental results fully demonstrate that in deep neural networks, multi-source transfer can effectively compensate for the lack of single-source transfer.

From Table 1, we see that our proposed MultiDTNN model is an iterative algorithm with a key parameter λ, so it is necessary to analyze its convergence and the influence of λ on the model. Below we analyze the convergence and the impact of parameters λ of MultiDTNN.

A. Convergence analysis

The training process of MulDTNN in Table 1 shows that the proposed algorithm consists of two sub-iterative processes: the first is that CNN is trained on source and target domains to obtain a set of classifier, and the other is to select classifiers from the set of classifier to compose an ensemble of classifier. Therefore, it is theoretically challenging to prove its convergence. So, we follow the researchers’ experience to obtain the convergence curve of our model as shown in Figure 4. As can be seen from Figure 4, our model has good convergence.

B. Parameter analysis

The parameter λ indicates the regularization coefficient in the objective function of MultiDTNN, which greatly affects the correlations between source and target domains. Therefore, we evaluate the influence of λ on model. Figure 5 gives a description of the classification performance over a range of three cross-domain tasks. We can see that MultiDTNN is a bell-shaped curve and can achieve better performance when the value is around 0.5. This also confirms that a good compromise between features of deep learning and distribution difference adaptation can enhance the transferability of features.

## 5. Conclusions

In this paper, we design a new deep transfer neural network framework: a multi-source deep transfer neural network, which integrates multi-source transfer learning, CNN, and JPDA into an optimization program. Multi-source transfer can provide more knowledge that is transferred into the target domain by using knowledge from multiple source domains, and the classification models of the target domain are built; CNN extracts more complex features of the dataset; JPDA is used to reduce the difference of probability distribution between domains and increases the transferability of features in source domains. Specifically, for the purpose of enhancing the transferability of features in deep neural networks, MultiDTNN utilizes JPDA to reduce the difference of domain probability distribution between each source and target domains. Then, on each source and target domains, we train CNN to obtain a set of deep learning classifiers. Finally, in order to select the classifier with the smallest classification error in the target domain from the classifier set, inspired by TaskTrAdaBoost a selection strategy is designed to obtain the MultiDTNN framework. The experimental results on the three cross-domain benchmark datasets demonstrate the effectiveness of our proposed model and have certain advantages over the benchmark algorithms. Although the experimental results show that the MultiDTNN has better classification performance than the benchmark algorithms, it still needs to work in the following aspects: further improve the convergence efficiency of the MultiDTNN model; in addition, it is also an interesting challenge to increase the number of source domains to more than 10.

## Figures and Tables

**Figure 1 sensors-19-03992-f001:**
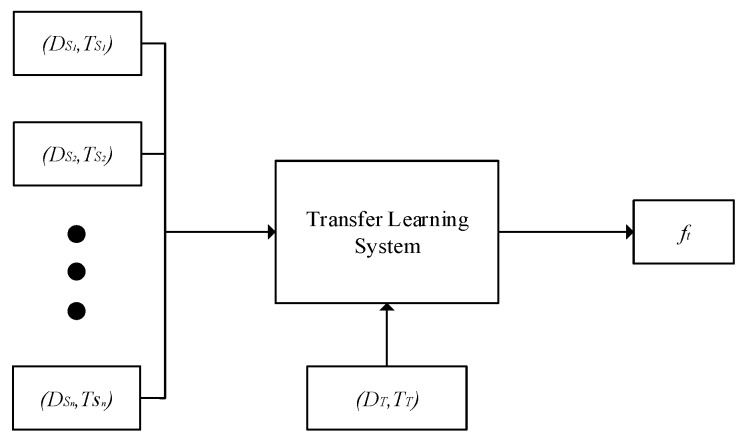
Framework of Multi-source Transfer Learning.

**Figure 2 sensors-19-03992-f002:**
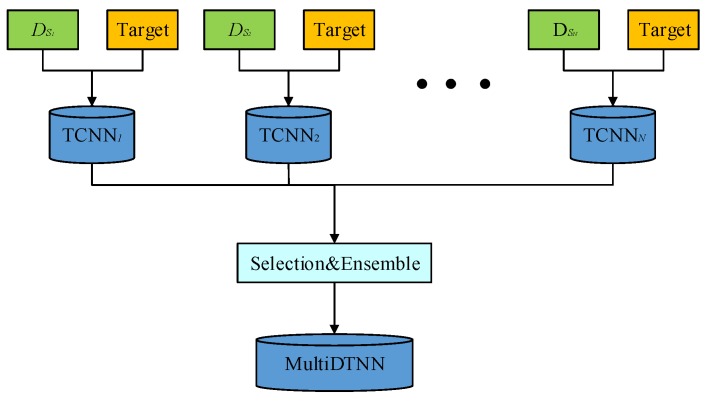
Framework of MultiDTNN.

**Figure 3 sensors-19-03992-f003:**
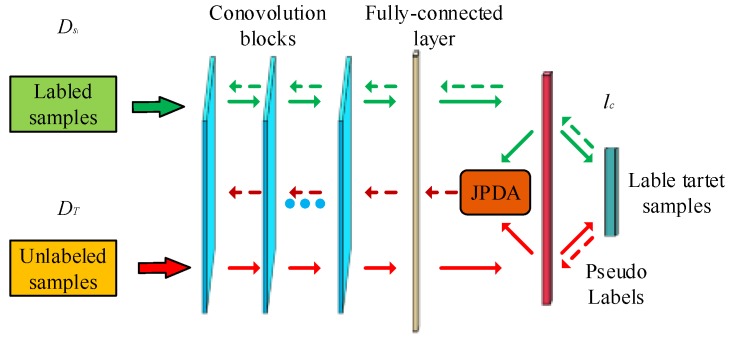
Structure of $TCNN_{s_{i}}$.

**Figure 4 sensors-19-03992-f004:**
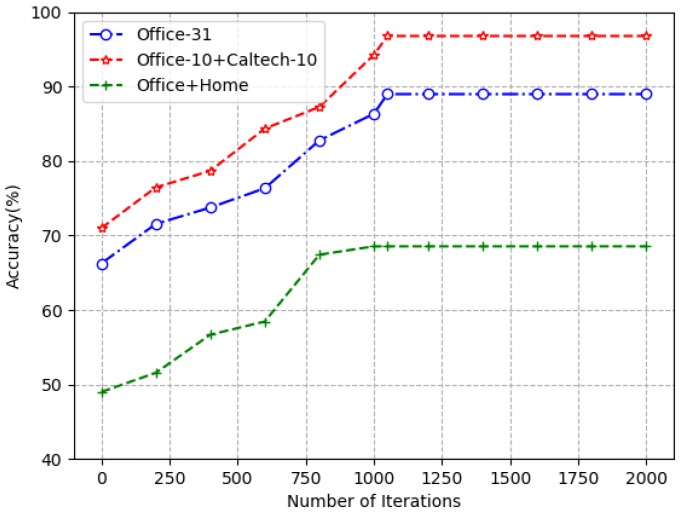
Converge curves of MultiDTNN on three datasets.

**Figure 5 sensors-19-03992-f005:**
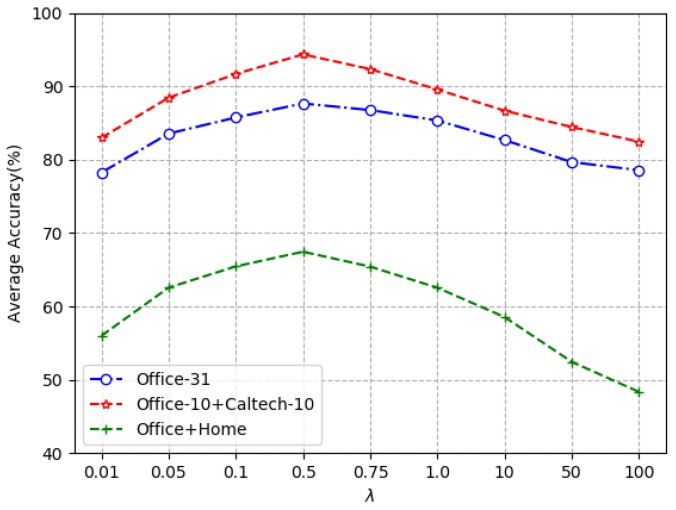
Influence of parameter λ on MultiDTNN on three datasets.

**Table 1 sensors-19-03992-t001:** Training Strategy of MultiDTNN.

**Training Procedure of** MultiDTNN
**Input:**N labeled source domains $D_{S}=\{D_{S_{i}}={\left(x_{j}^{s_{i}},y_{j}^{s_{i}}\right)}_{j=1}^{n_{s_{i}}},i=1,\ldots ,N\}$, the number of sample in $s_{i}$ is $n_{s_{i}}$. An unlabeled training dataset $D_{T}={\left(x_{i}\right)}_{i=1,\ldots ,n_{t}}$ in target domain, the architecture of deep neural network, the trade-off parameter λ, the maximum number of iterations M. **Output:** $\hat{f_{T}}$ **Training:** Step 1. Pre-train a set of classifier $H\in \{TCNN_{1},TCNN_{2},\ldots ,TCNN_{N}\}$ on $D_{T}\cup D_{S}$; H←∅ for i←0 to N do Train base deep network $CNN_{i}$ on $D_{s_{i}}$ Predict the pseudo labels ${\hat{Y}}_{0}={\left(y_{k}^{t}\right)}_{k=1}^{n_{t}}$ on $D_{T}$ by using $CNN_{i}$ Repeat j=j+1 Compute the regularization term JPDA according to Equation (9) Obtain $TCNN_{i}$ by optimizing $CNN_{i}$ with Equation (10) Update the pseudo labels ${\hat{Y}}_{j}$ with optimized network $TCNN_{i}$ Until convergence or ${\hat{Y}}_{j}={\hat{Y}}_{j-1}$, $H\leftarrow H\cup TCNN_{i}$ Step 2. Initialize the weight vector $w^{T}=(w_{1}^{T},w_{2}^{T},\ldots ,w_{n_{t}}^{T})$ for t←0 to M do Step 3. The weight vector $w^{T}$ are normalized to 1 Step 4. Empty the current weak classifier set F←∅ for t←0 to N do Step 5. Compute the error $\varepsilon_{t}$ of $h^{k}\in H$ on $D_{T}$ $\varepsilon_{t}\leftarrow \sum_{j}w_{j}^{T}[y_{j}^{T}\neq h^{k}(x_{j}^{T})]$ if $\varepsilon_{t}>1/2$ then Step 6. $h^{k}\leftarrow -h^{k}$ Step 7. Update by compute $\varepsilon_{t}\leftarrow \sum_{j}w_{j}^{T}[y_{j}^{T}\neq h^{k}(x_{j}^{T})]$ Step 8. $F\leftarrow F\cup (h^{k},\varepsilon_{t})$ Step 9. Find the weak classifier $h_{t}:x\to y$ $(h_{t},\varepsilon_{t})=\arg\underset{(h^{k},\varepsilon_{t})\in F}{\min\varepsilon_{t}}$ Step 10. $H\leftarrow H-h_{t}$ Step 11. Set $\alpha_{t}=\frac{1}{2}\ln\frac{1-\varepsilon_{t}}{\varepsilon_{t}}$ Step 12. Update the weight $w_{i}^{T}\leftarrow w_{i}^{T}e^{-\alpha_{t}y_{i}^{T}h_{t}^{k}(x_{i}^{T})}$ Step 13. Return $\hat{f_{T}}=\mathrm{sign}(\sum_{t}\alpha_{t}h_{t}(x))$

**Table 2 sensors-19-03992-t002:** Average accuracy rate (%) with absolute value of standard variation on Office-31 dataset.

Algorithms	A->W	D->W	A->D	W->D	D->A	W->A
CNN [38]	60.15 (0.45)	94.33 (0.35)	63.16 (0.46)	98.23 (0.19)	50.98 (0.58)	50.01 (0.38)
DTLC [29]	70.78 (0.31)	97.11 (0.56)	68.67 (0.52)	99.23 (0.36)	55.56 (0.32)	54.11 (0.56)
STLCF [16]	58.11 (0.39)	92.26 (0.41)	60.87 (0.36)	96.14 (0.26)	48.98 (0.45)	48.87 (0.43)
DAN [31]	69.52 (0.43)	95.96 (0.34)	67.14 (0.42)	99.01 (0.21)	54.23 (0.37)	53.23 (0.34)
SDT [32]	67.78 (0.32)	96.12 (0.43)	66.57 (0.52)	98.86 (0.38)	54.45 (0.23)	54.12 (0.37)
DHN [34]	68.27 (0.43)	96.15 (0.23)	66.55 (0.28)	98.56 (0.33)	55.97 (0.27)	52.65 (0.23)
ARTL [12]	57.27 (0.51)	93.57 (0.37)	59.31 (0.45)	95.45 (0.22)	49.14 (0.43)	47.36 (0.33)
D-CORAL [40]	67.24 (0.37)	95.68 (0.35)	66.87 (0.58)	99.23 (0.26)	52.35 (0.34)	51.26 (0.32)
DDC [35]	62.02 (0.46)	95.02 (0.53)	65.23 (0.39)	98.43 (0.41)	52.13 (0.67)	51.98 (0.46)
	{A,W,D}->W		{A,W,D}->D		{A,W,D}->A	
TaskTrBoost [22]	66.67 (0.42)	94.67 (0.48)	64.76 (0.35)	95.67 (0.43)	51.34 (0.38)	50.24 (0.34)
FastDAM [24]	68.34 (0.37)	95.86 (0.46)	65.32 (0.38)	98.43 (0.35)	52.72 (0.42)	52.36 (0.32)
IMTL [26]	70.45 (0.35)	96.98 (0.51)	66.15 (0.43)	99.11 (0.36)	53.98 (0.53)	53.65 (0.41)
MultiDTNN	73.65 (0.29)	98.13 (0.52)	70.01 (0.43)	99.56 (0.38)	57.11 (0.54)	56.98 (0.35)

**Table 3 sensors-19-03992-t003:** Average accuracy rate (%) with absolute value of standard variation on Office-10+Caltech-10 dataset.

Algorithms	A->C	D->C	W->C	A->W	C->W	D->W	A->D	C->D	W->D	C->A	D->A	W->A
CNN [38]	83.76 (0.33)	81.23 (0.43)	75.89 (0.55)	83.24 (0.29)	82.87 (0.35)	97.53 (0.24)	88.65 (0.36)	89.34 (0.31)	98.14 (0.27)	91.01 (0.23)	89.23 (0.28)	83.25 (0.32)
DTLC [29]	88.76 (0.65)	83.23 (0.34)	82.45 (0.41)	93.56 (0.48)	93.01 (0.58)	99.54 (0.31)	93.77 (0.43)	91.39 (0.36)	99.47 (0.13)	93.46 (0.62)	93.18 (0.52)	94.12 (0.59)
STLCF [16]	82.25 (0.43)	80.56 (0.56)	74.32 (0.58)	82.11 (0.53)	81.22 (0.51)	96.34 (0.52)	87.34 (0.42)	88.34 (0.48)	97.21 (0.24)	89.53 (0.43)	88.43 (0.51)	82.13 (0.47)
DAN [31]	86.01 (0.28)	82.56 (0.38)	81.62 (0.36)	93.88 (0.43)	92.12 (0.37)	99.11 (0.22)	92.16 (0.29)	90.83 (0.27)	99.12 (0.11)	91.87 (0.31)	92.11 (0.48)	92.46 (0.35)
SDT [32]	85.24 (0.32)	81.98 (0.45)	80.87 (0.36)	93.67 (0.34)	92.11 (0.54)	99.26 (0.41)	91.58 (0.46)	90.45 (0.37)	99.32 (0.18)	91.12 (0.31)	91.34 (0.43)	91.42 (0.48)
DHN [34]	86.35 (0.26)	82.12 (0.42)	81.23 (0.32)	93.32 (0.25)	92.45 (0.21)	99.15 (0.37)	89.57 (0.31)	90.11 (0.35)	99.26 (0.19)	92.11 (0.28)	91.68 (0.36)	91.63 (0.34)
ARTL [12]	81.46 (0.43)	79.26 (0.43)	73.87 (0.49)	81.87 (0.55)	80.87 (0.52)	95.23 (0.56)	86.32 (0.48)	87.96 (0.47)	98.43 (0.28)	88.41 (0.38)	88.01 (0.44)	81.97 (0.54)
D-CORAL [40]	85.87 (0.37)	82.45 (0.28)	81.34 (0.22)	92.59 (0.32)	91.37 (0.38)	99.34 (0.31)	89.26 (0.35)	89.98 (0.42)	99.56 (0.15)	92.43 (0.33)	91.76 (0.37)	91.64 (0.24)
DDC [35]	84.23 (0.52)	81.26 (0.37)	78.13 (0.53)	86.54 (0.41)	82.15 (0.43)	98.26 (0.38)	89.11 (0.38)	89.74 (0.45)	99.67 (0.21)	92.21 (0.35)	90.12 (0.42)	85.15 (0.47)
	{A,D,W,C}->C			{A,D,W,C}->W			{A,D,W,C}->D			{A,D,W,C}->A		
TaskTrBoost [22]	83.56 (0.41)	81.64 (0.34)	80.26 (0.53)	88.34 (0.52)	87.45 (0.47)	97.33 (0.46)	88.35 (0.51)	89.56 (0.43)	97.78 (0.24)	91.25 (0.33)	89.23 (0.38)	88.67 (0.36)
FastDAM [24]	84.32 (0.35)	82.11 (0.45)	81.23 (0.47)	90.32 (0.48)	89.21 (0.43)	98.32 (0.49)	89.35 (0.46)	90.65 (0.39)	98.34 (0.29)	92.56 (0.29)	92.27 (0.42)	92.35 (0.48)
IMTL [26]	85.77 (0.43)	83.43 (0.36)	82.15 (0.52)	92.61 (0.46)	91.23 (0.53)	99.65 (0.44)	92.36 (0.51)	91.01 (0.36)	99.45 (0.27)	93.79 (2.31)	93.44 (0.45)	94.86 (0.52)
MultiDTNN	89.34 (0.53)	85.64 (0.31)	84.58 (0.43)	94.78 (0.42)	94.55 (0.44)	99.96 (0.37)	94.38 (0.46)	92.24 (0.32)	99.98 (0.14)	94.15 (0.27)	95.01 (0.48)	95.28 (0.53)

**Table 4 sensors-19-03992-t004:** Average accuracy rate (%) with absolute value of standard variation on Office+Home dataset.

Algorithms	Ar->Cl	Pr->Cl	Rw->Cl	Ar->Pr	Rw->Pr	Cl->Pr	Ar->Rw	Cl->R	Pr->Rw	Cl->Ar	Pr->Ar	Rw->Ar
CNN [38]	30.11 (0.56)	34.56 (0.37)	38.72 (0.38)	39.23 (0.54)	60.32 (0.33)	46.76 (0.46)	50.23 (0.45)	49.54 (0.39)	54.32 (0.46)	32.25 (0.53)	28.45 (0.41)	42.54 (0.65)
DTLC [29]	35.53 (0.65)	41.57 (0.36)	44.62 (0.43)	43.76 (0.45)	66.11 (0.32)	52.89 (0.69)	56.32 (0.54)	53.54 (0.35)	61.56 (0.51)	36.79 (0.53)	32.35 (0.48)	45.75 (0.33)
STLCF [16]	29.43 (0.47)	33.67 (0.43)	37.65 (0.34)	38.55 (0.51)	59.87 (0.43)	45.78 (0.55)	49.45 (0.53)	48.43 (0.47)	53.76 (0.48)	31.34 (0.42)	27.54 (0.54)	41.65 (0.53)
DAN [31]	30.32 (0.54)	34.16 (0.32)	38.45 (0.42)	42.56 (0.51)	62.76 (0.28)	47.65 (0.55)	54.27 (0.58)	50.12 (0.33)	56.82 (0.45)	32.67 (0.49)	30.11 (0.45)	43.67 (0.31)
SDT [32]	32.65 (0.42)	35.87 (0.54)	42.55 (0.34)	41.32 (0.48)	64.45 (0.33)	49.56 (0.46)	52.77 (0.53)	50.56 (0.35)	54.82 (0.48)	33.67 (0.34)	30.67 (0.43)	43.45 (0.29)
DHN [34]	31.75 (0.44)	40.13 (0.37)	45.23 (0.41)	40.85 (0.56)	62.89 (0.36)	52.01 (0.58)	51.75 (0.55)	52.82 (0.46)	61.23 (0.53)	34.78 (0.35)	31.24 (0.43)	45.23 (0.38)
ARTL [12]	28.23 (0.57)	32.74 (0.53)	36.66 (0.38)	37.12 (0.41)	58.58 (0.47)	44.27 (0.42)	48.87 (0.54)	47.84 (0.56)	52.57 (0.44)	30.13 (0.52)	26.62 (0.51)	40.54 (0.43)
D-CORAL [40]	30.85 (0.43)	34.28 (0.38)	40.35 (0.39)	42.34 (0.53)	62.56 (0.45)	47.26 (0.61)	54.56 (0.52)	48.87 (0.41)	55.67 (0.48)	32.67 (0.46)	28.75 (0.44)	43.81 (0.42)
DDC [35]	31.25 (0.56)	36.52 (0.31)	39.65 (0.37)	41.87 (0.43)	63.65 (0.53)	48.54 (0.55)	53.56 (0.48)	51.67 (0.32)	57.31 (0.44)	31.82 (0.36)	29.67 (0.46)	44.78 (0.38)
	{Ar,Pr,Rw,Cl}->Cl			{Ar,Pr,Rw,Cl}->Pr			{Ar,Pr,Rw,Cl}->Rw			{Ar,Pr,Rw,Cl}->Ar		
TaskTrBoost [22]	30.32 (0.46)	34.26 (0.34)	38.46 (0.43)	39.35 (0.46)	61.88 (0.37)	49.87 (0.51)	51.86 (0.55)	49.56 (0.39)	56.43 (0.51)	32.65 (0.31)	29.54 (0.53)	42.34 (0.32)
FastDAM [24]	31.54 (0.43)	35.65 (0.37)	41.87 (0.39)	41.23 (0.42)	62.44 (0.41)	50.54 (0.48)	53.25 (0.57)	51.28 (0.36)	59.53 (0.53)	34.43 (0.44)	30.23 (0.58)	43.11 (0.36)
IMTL [26]	32.24 (0.47)	36.54 (0.33)	43.32 (0.48)	43.45 (0.38)	64.56 (0.45)	52.37 (0.52)	55.11 (0.49)	52.21 (0.34)	61.88 (0.55)	35.98 (0.46)	32.11 (0.51)	45.21 (0.43)
MultiDTNN	36.88 (0.44)	41.34 (0.28)	46.21 (0.45)	45.45 (0.42)	68.65 (0.43)	53.56 (0.54)	57.63 (0.51)	55.32 (0.37)	63.27 (0.57)	38.23 (0.34)	34.24 (0.56)	46.34 (0.44)

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
