# Peer review of "Multi-Source Deep Transfer Neural Network Algorithm"

_sensors, 2019, doi:10.3390/s19183992_

Round 1

Reviewer 1 Report

My comments and questions are given in the attached file.

Reviewer 2 Report

In Figure 2, there is the process called “Selection&Ensemble”. Is it based on the features, layers or models (like voting) ensemble? The authors should make it clear. AlexNet structure is adopted in the system. While the performance of the ResNet is determined by numbers of layers somehow. AlexNet structure is relatively short for ResNet. The authors have addressed more on the transfer learning system and the definition of multisource deep models, while discussed less about the specific transfer learning method. The authors should introduce transfer learning in Section 3. Format and language:

(1) The order of references is confusing, especially in the first paragraph.

(2) In Section 2.2, the line 5, [38][39] is usually is expressed as [38, 39].

(3) In Equation (1), italic is not necessary for ReLU.

(4) The authors should show that joint probability distribution adaptation is JPDA.

(5) If there are no supplementary materials, the authors should delete Lines 431-432.

Round 2

Reviewer 1 Report

All my comments and questions have been addressed and clarified. The manuscript has been improved and now warrants publication.

This manuscript is a resubmission of an earlier submission. The following is a list of the peer review reports and author responses from that submission.

Round 1

Reviewer 1 Report

In this paper authors are summarizing their work on building a multi-source deep transfer neural network. The contribution of the paper is not sufficient for publication. Actually, this work is not within the current scope of the journal Sensors.   The primary claims of the paper are as follow: a. the deep transfer structures are constructed based on joint probability distribution and convolutional neural network. b. more knowledge in the multi-source domains are provided to assist in building the learning model of target domain. c. ensemble system of classifiers is more advantageous than a single classifier in terms of prediction effectiveness and stability.   For claim (a), the deep networks with joint distribution adaptation method is first proposed by Mingsheng Long in ICML2017, which is famous as JAN(joint adaptation networks). Therefore, this claim is not a promising contribution. For claim (b), the authors proposed to learn TCNN for each source respectively and select the final model with existing adaboost-based method. Therefore, the proposed framework is the combination of existing methods. For claim (c), the comparison is not sufficient to demonstrate its utility as a multi-source learning method. The benchmark in the paper only use one source, while the proposed method use three source. The author should compare the proposed method with other multi-source transfer learning method. In addition, the abbreviation of the proposed method is inconsistency, MultiDTNN or MultiTLGP?

Reviewer 2 Report

My comments and questions are given in the attached file.
